# Pathogenesis and Personalized Interventions for Pharmacological Treatment-Resistant Neuropsychiatric Symptoms in Alzheimer’s Disease

**DOI:** 10.3390/jpm12091365

**Published:** 2022-08-24

**Authors:** Tomoyuki Nagata, Shunichiro Shinagawa, Keisuke Inamura, Masahiro Shigeta

**Affiliations:** 1Department of Psychiatry, The Jikei University School of Medicine, Tokyo 105-8461, Japan; 2Center for Dementia-Related Diseases, Airanomori Hospital, Kagoshima 890-0065, Japan

**Keywords:** Alzheimer’s disease, neuropsychiatric symptoms (NPSs), treatment-resistant, antipsychotics, non-pharmacological symptom, the Clinical Antipsychotic Trials of Intervention Effectiveness–Alzheimer’s Disease (CATIE-AD) trial

## Abstract

Alzheimer’s disease (AD) is the most common form of dementia, with cognitive impairment as a core symptom. Neuropsychiatric symptoms (NPSs) also occur as non-cognitive symptoms during the disease course, worsening the prognosis. Recent treatment guidelines for NPSs have recommended non-pharmacological treatments as the first line of therapy, followed by pharmacological treatments. However, pharmacological treatment for urgent NPSs can be difficult because of a lack of efficacy or an intolerance, requiring multiple changes in psychotropic prescriptions. One biological factor that might be partly responsible for this difficulty is structural deterioration in elderly people with dementia, which may cause a functional vulnerability affecting the pharmacological response. Other causative factors might include awkward psychosocial interpersonal relations between patients and their caregiver, resulting in distressful vicious circles. Overlapping NPS sub-symptoms can also blur the prioritization of targeted symptoms. Furthermore, consistent neurocognitive reductions cause a primary apathy state and a secondary distorted ideation or perception of present objects, leading to reactions that cannot be treated pharmacologically. The present review defines treatment-resistant NPSs in AD; it may be necessary and helpful for clinicians to discuss the pathogenesis and comprehensive solutions based on three major hypothetical pathophysiological viewpoints: (1) biology, (2) psychosociology, and (3) neurocognition.

## 1. Introduction

### 1.1. Neuropsychiatric Symptoms in Alzheimer’s Disease

By 2050, the prevalence of patients with dementia worldwide is expected to increase explosively, reaching about 152 million [1]. Alzheimer’s disease (AD) is the most common form of dementia and is characterized by progressive neurocognitive impairments (e.g., memory impairment, executive dysfunction, and visuospatial deficits), leading to reductions in the activities of daily living (ADLs) [2]. Various neuropsychiatric symptoms (NPSs), including psychosis, agitation, depression, wandering, aberrant motor behaviors, and apathy, appear during the course of dementia [3]. NPSs in patients with AD are associated with a more rapid disease progression and increased mortality risk [4,5]. Especially, severe psychosis and agitation can also increase the direct costs associated with the use of healthcare as well as indirect costs arising from the economic burden, such as time away from work or leisure activities [6,7,8,9]. Collectively, NPSs worsen patients’ prognoses, resulting in (1) a shorter life span; (2) earlier institutionalization; (3) more severe caregiver burden; and (4) increased socio-economic burden; thus, provisional solutions for NPSs, including both pharmacological and non-pharmacological interventions, are an urgent issue [3,4,5,6,7,8,9].

### 1.2. Trends in Pharmacological Treatments for NPSs and the Resulting Dilemma

Several major associations or societies have prepared clinical treatment guidelines for NPSs in dementia that recommend non-pharmacological interventions (e.g., music therapy, reminiscence, and behavioral management techniques) rather than pharmacological interventions [10,11,12]. The treatment algorithm for NPSs in patients with dementia requires the existence of the following definitive, urgent conditions for pharmacological treatment: (1) major depression with or without suicidal ideation; (2) psychosis causing harm or with great potential for harm; and (3) agitation causing risk to self or others [11]. The guidelines also emphasize the necessity of monitoring for adverse effects caused by psychotropics and the use of pharmacological treatments for a short duration after non-pharmacological interventions [11,12,13].

Antipsychotics have conventionally been presumed to be more effective than other medicines for improving psychosis or agitation in patients with AD; however, the reproducibility of such results is unclear [11,12,13]. The Clinical Antipsychotic Trials of Intervention Effectiveness–Alzheimer’s Disease (CATIE-AD) were performed from 2000 to 2004. This 36-week, large-scale, double-blind, placebo-controlled study investigated the longitudinal effectiveness and safety of atypical antipsychotics (AAP) for the treatment of mainly psychotic or aggressive symptoms in 421 patients with AD. [14,15]. However, the CATIE-AD trial could not show the significant effectiveness of AAPs, compared with a placebo, and about 80% (*n* = 223) of patients who had been prescribed an active drug (risperidone, olanzapine, or quetiapine; *n* = 279) discontinued treatment during phase 1 (first assigned drug; see Figure 1) [15]. The main reasons for discontinuation were undesirable adverse effects (19%; *n* = 54) and lack of efficacy (45%: *n* = 126), revealing the poor effectiveness and intolerance of first-assigned AAPs against NPSs (Figure 1) [15]. Furthermore, the CATIE-AD trial has shown that 25% (*n* = 69) of all subjects with any AAPs needed to perform at least a second switching of active drugs within 36 weeks (Figure 1). These results suggest that the treatment of NPSs is difficult to complete using a first-choice AAP, implying that about one-fourth of patients with AD who are treated with an AAP will need to switch the prescribed AAP multiple times. When considering alternative medicines for pharmacological treatment-resistant psychotic or aggressive symptoms, a confirmed treatment strategy for patients with schizophrenia, which has already been standardized in a treatment guideline [16], could be considered. While the effectiveness of clozapine has been described in the American Psychiatric Association (APA) guidelines as an alternative medicine for patients with treatment-resistant schizophrenia at risk of suicide or with aggressive behaviors, clozapine has been used in elderly patients infrequently because of undesirable adverse effects including agranulocytosis, metabolic side effects, and myocarditis [16,17]. Therefore, as a pharmacological alternative strategy, the usage of clozapine for NPS in elderly people with dementia may be not recommendable, except for treating psychosis in Parkinson’s disease or dementia with Lewy bodies (DLB) [16,17,18]. Among the representative NPS sub-symptoms in patients with AD mentioned above, depression is considered to be an urgent sub-symptom, and pharmacological treatments have been suggested in a guideline [11]. However, the evidence concerning antidepressant treatment for depression in AD has been suggested to be inconclusive by some narrative reviews and meta-analyses seeking to identify the efficacy and safety of such treatments [19,20,21].

The crucial sub-symptoms (psychosis, agitation, and depression) of NPSs that necessitate urgent pharmacological interventions tend to be difficult to treat, with the first-choice medicine often having to be switched to another drug. However, non-pharmacological interventions (e.g., music therapy, reminiscence, and behavioral management techniques) for the treatment of NPSs may be relatively weak alternatives to pharmacological treatments, since psychosis, agitation, and depression can result in harmful behaviors to the patient and to others, and require urgent treatment to prevent secondary misfortune [22]. Therefore, elucidating the pathogeneses of NPSs that are difficult to treat pharmacologically may be helpful to clinicians and caregivers during discussions regarding long-term care plans and treatment strategies, possibly improving patient prognosis, including the quality of life (QOL), and mortality in patients with AD [3,4,5].

### 1.3. Scope of the Present Review

Biological or psychosocial factors have already been investigated as conventional causative factors of various NPS sub-symptoms from a comparatively short-term perspective [14,22]. Furthermore, previous reviews and studies have suggested that a neurocognitive viewpoint might also be needed as a crucial “third causative factor” with relevance to the longitudinal prognosis [4,5,22,23]. Some sub-symptoms based on secondary reactions originating from consistent deterioration (neurocognitive reductions) may have poorer responses to psychotropic treatments [22]. Focusing on associations with neurocognition in AD may contribute to the elucidation of the mechanisms responsible for ideational, perceptual, or motivational alterations of present objects, which could lead to the resolution of treatment-resistant NPSs including delusional misidentification (DMS) and apathy [22]. Therefore, the following three viewpoints—biology, psychosociology, and neurocognition—should be discussed to resolve these unknown mechanisms using a bio-psychosocial-neurocognitive model. The present narrative review will consider the complex hypothetical pathogenesis of symptoms that are difficult to treat pharmacologically from these three viewpoints, and will discuss future perspectives regarding treatment strategies and solutions for these elusive issues.

## 2. Definition of Pharmacological Treatment-Resistant NPSs

While pharmacological treatment difficulties in patients with mental disorders have been discussed and defined in previous reports, those concerning NPSs in patients with dementia have not yet been described in detail [24,25,26]. During long-term pharmacological treatment for common mental disorders including schizophrenia and depression, a certain proportion of patients respond poorly to pharmacological treatment. The definition of treatment-resistant schizophrenia (TRS) has been discussed in several previous reports, and the conclusion seems to be that treatment failure can occur despite the provision of adequate treatment including the specified minimum dosage, duration, and number of previous antipsychotics taken [24,25]. The Treatment Response and Resistance in Psychosis (TRRIP) Working Group discussed and identified the optimal criteria for TRS: patients with schizophrenia with (1) at least moderate severity and less than 20% symptom reductions after 6 weeks or longer of treatment with a therapeutic antipsychotic dosage (equivalent to 600 mg or more of chlorpromazine per day), (2) a duration of treatment resistance of 12 weeks or more, and (3) at least two different treatments with antipsychotic drugs and at least one use of a long-acting injectable antipsychotic [25]. Meanwhile, the definition of pharmacological treatment-resistant depression (TRD) was summarized as the failure to achieve a treatment response after at least two consecutive, adequate antidepressant trials of adequate treatment duration [26]. Adequate treatment duration means an antidepressant trial lasting for at least 4 weeks and with an optimal dose of antidepressant [26]. Concerning the changes in severity, the Hamilton Depression Rating Scale (HAMD-17) score should not decrease to 17 points or fewer despite the use of at least 2 adequate consecutive antidepressants during the last episode [26]. Furthermore, treatment-resistant bipolar depression (TRBD) was defined as a lack of response after at least two consecutive adequate antidepressant treatments in combination with adequate treatment with an established mood stabilizer [26]. As described above, some previous reports have regarded clinical “treatment resistance” in patients with mental disorders to result from a poor response against adequate pharmacological treatments, rather than non-pharmacological interventions [22,24,25,26].

Among neuropsychiatric sub-symptoms in AD, conditions such as psychosis, agitation, or depression may require pharmacological treatments similar to those used for schizophrenia and major depressive disorder; however, symptoms that are typically treated pharmacologically may sometimes lead to “treatment resistance” if a poor response or intolerability occurs despite adequate medication. Therefore, in the present review, we advocate that “pharmacological treatment-resistant NPSs in AD (p-TRENS-AD)” should be used to describe urgent sub-symptoms (psychosis, aggressiveness, and depression) that are likely to respond poorly to pharmacological treatment or for which pharmacological treatment is likely to be intolerable to the patient, resulting in a need to switch to multiple psychotropics. The effectiveness of pharmacological treatment has been discussed in previous studies, but sub-symptoms such as wandering, perseverative shouting, and some sexually inappropriate behaviors are unlikely to respond to a pharmacological approach [11,22,27]. Thus, a pharmacological approach might not be suitable for the treatment of these sub-symptoms when they first appear. The p-TRENS-AD overlapping with such sub-symptoms (wandering, perseverative shouting, and inappropriate behaviors) should be defined as a solution to a difficult issue.

## 3. Hypothetical Pathogenesis of p-TRENS-AD

Previous guidelines, meta-analyses, and literature reviews have discussed the effectiveness and necessity of pharmacological treatments (antipsychotics, antidepressants, anticonvulsants, and antidementia drugs) for NPS sub-symptoms (psychosis, agitation, and depression) in patients with dementia, and the results of these studies are summarized in Table 1 [13,19,20,21,22,28,29]. As mentioned above, two conventional categories of causative factors are biological (factors that influence the pharmacological response or tolerability during the treatment period) and psychosocial (factors causing interpersonal conflict with cohabitants in daily life regardless of the response to psychotropics). Furthermore, a consistent neurocognitive reduction itself can be included as a complicating factor in NPSs, and such neurocognitive reductions may cause either primary problems, such as apathy because of disease progression, or secondary problems, such as reactions resulting from ideational and perceptional alterations (Table 2).

Since these urgent sub-symptoms requiring psychotropic usage often cause treatment difficulties, a theoretical discussion based on the above three viewpoints is likely to be noteworthy [22].

### 3.1. Biological Factors: Pharmacokinetic, Metabolic, Pharmacogenomic, and Neuropharmacology

The discontinuation of pharmacological treatment in AD is influenced by undesirable adverse effects and a lack of efficacy, and treatment must often be switched to an alternative medication [14,15]. Generally, elderly people may have difficulty tolerating pharmacological treatments, and an initial approach to dosing guided by “start low, go slow” is needed to reduce the risk of adverse effects [30]. Age-related alterations in peripheral pharmacokinetics reduce drug clearance, contributing to undesirable adverse effects arising from elevated blood concentrations; however, this phenomenon is not consistent for antipsychotics [31,32]. For the pharmacological management of schizophrenia, a certain optimal occupancy (60–80%) of striatal dopamine D2/3 receptors is regarded as the “therapeutic window” for the prescription of antipsychotics, based on positron emission tomography (PET) neuroreceptor imaging studies examining the clinical response range in patients with minimal extrapyramidal symptoms (EPS) [33,34]. In previous studies, however, older patients with schizophrenia were shown to have a lower therapeutic window (50–60% D2/3 relative receptor occupancy) than younger patients [35,36]. Another study showed that a lower antipsychotic (amisulpride) dosage was capable of achieving the therapeutic window of D2/3 receptor occupancy in patients with AD who were being treated for psychosis [37]. Within the AD-affected brain, changes in the central drug absorption system arising from the disruption of the blood–brain barrier (BBB) may increase the affinity of antipsychotics to the D2/3 receptor, leading to increased permeability across the BBB and a reduction in efflux transporters [37,38]. Furthermore, cytochrome P450 (CYP) isoforms (mainly 2D6 and 3A4) may exacerbate undesirable effects via drug–drug interactions (DDI) arising from the concomitant use of psychotropics and multiple other medicines prescribed for lifestyle-related disease [13,39]. Alterations in the pharmacokinetic system in elderly subjects with AD may result in a lack of treatment efficacy, or intolerance to antipsychotics typically prescribed for the treatment of psychosis and agitation, leading to treatment discontinuation [14,15].

From the viewpoint of pharmacogenomics, the antidepressant response in patients with major depressive disorder (MDD) was significantly predicted by candidate gene polymorphisms (single nucleotide polymorphism: SNPs), such as those in catecholamine-related enzymes or receptors, neurotrophic factors, and transcription factors (e.g., *COMT*; *5HTR2A*; *BDNF*; *CREB1*), but was not associated with CYP genes (*CYP1A2*; *CYP2C9*; *CYP2C19*; *CYP2D6*) [26]. Especially, a significant association between the *COMT* gene and suicidal behaviors in MDD patients who did not respond to antidepressant treatment has been found in a previous study [40]. Meanwhile, a better response to treatment for depression and anxiety in patients with AD has been associated with *APOE*-ε3 carriers [41]. The metabolic genophenotypes of the CYP enzymes *CYP2D6*, *CYP2C19*, and *CYP2C9* can be classified as normal metabolizer (NM), intermediate metabolizer (IM), poor metabolizer (PM), or ultra-rapid metabolizer (UM), and patients with AD who were classified as NMs or IMs responded significantly better to current treatments for depression and anxiety than those who were classified as PMs or UMs [41]. Such results imply that the severity of depressive symptoms needing pharmacological treatment, including suicidal attempts, may be associated with a genomic predisposition (e.g., *COMT*, *APOE*, *CYP*), leading to a poorer response or a treatment-resistant status via the metabolic system [41,42]. However, the previous discrepancies in antidepressant efficacy between patients with MDD and AD patients with similar depressive symptom cannot be explained by this pharmacogenomic paradigm alone; thus, other factors (psychosocial or neurocognitive) may influence the subsequent prognosis in AD patients with depression [19,20,21].

The theoretical neuropharmacological mechanism of psychosis in AD is subtly different from that for schizophrenia mainly because of the relevance of the monoaminergic neurocircuitry, since basal forebrain cholinergic loss might be linked to the reciprocal dopaminergic activation associated with psychosis [42,43,44,45]. While the progressive loss of cholinergic innervation contributes to the main cognitive impairments in AD, the loss of dopaminergic neurons in the substantia nigra also progresses simultaneously throughout the course of disease degeneration [43,44,45,46]. We speculated that these two paradoxical neuropharmacological alterations of reciprocal cholinergic-dopaminergic neurocircuitry, i.e., the relative striatal hyperdopaminergic state and dopaminergic neuronal loss within the AD brain, may make it difficult for neuropharmacological treatment using antipsychotics to result in sensitive EPS responses, causing a phenomenon that cannot be explained by pharmacokinetic effects only. Depression and aggression may also originate from the involvement of monoaminergic neurons, including the serotonergic dorsal raphe nucleus (DRS) and the noradrenergic locus coeruleus (LC). This might be relevant to early tau protein abnormalities, suggesting subcortical neurofibrillary tangle accumulation (Figure 2) [45,46]. The DRS or LC is also involved in the control of the sleep-wake cycle, which means that its impairment may cause diurnal (circadian) rhythm disorders including wake-sleep cycle disturbances in patients with AD [45,46]. As a reason why patients with late-life depression (LLD) have a more heterogeneous response to antidepressants than patients with mid-life depression, serotonergic system degeneration and a variability in serotonergic transporter occupancy may contribute to the treatment response in patients with LLD [47]. Likewise, such variable responses to the treatment of depression in patients with AD may also be observed for antidepressant therapy [19,20,21,48]. Despite the inconsistent results in the response of AD patients to the treatment of depression, recent studies have shown the efficacy of a selective serotonin reuptake inhibitor (SSRI) or a serotonin noradrenalin reuptake inhibitor (SNRI) for the treatment of aggressiveness in AD, which may support the consistent compensatory activation of surviving noradrenergic projections [22,49,50]. The decreases in cholinergic function in AD are significantly correlated with aggressive behavior, suggesting a reciprocal role of the cholinergic systems via monoaminergic neuronal activation [51]. Therefore, dopamine-2 receptor blocker agents like antipsychotics may be effective against the monoaminergic hyper-availability state leading to delusion and aggressiveness, but they may tend to accelerate apathy or motivation loss arising from the hypoactive state of the monoaminergic or cholinergic systems [42,43,45,46]. In addition to functional loss in the cholinergic system, a relatively active state of monoaminergic neurocircuit alterations may contribute to various p-TRENS-AD, including psychosis, aggressiveness, and depression.

Collectively, the above findings suggest that pharmacokinetic alterations arising from the therapeutic window within the brain, pharmacogenomic phenotypes, and monoaminergic-cholinergic system alterations may influence the unstable treatment response to selected antipsychotics or antidepressants in elderly people with AD, leading to treatment discontinuation or a need to switch prescriptions.

### 3.2. Psychosocial Viewpoints: Distressful Vicious Circle between Caregiver and Patient, Multiple Demographic Factors, and Comorbid Sub-Symptoms

Regarding psychosocial or demographic factors for NPSs (e.g., delusion, agitation, and depression), severe caregiver burden, housemate type, racial type, educational level, sex, marital status, residential situation, and comorbidities other than NPS sub-symptoms have been associated with NPS emergence [52,53,54,55,56,57,58,59]. The reciprocal relationship between psychosocial factors and demographic ones may influence the occurrence of NPSs in AD, causing interpersonal problems between patients and caregivers via psychopathological conflicts [22]. For instance, among the predominant factors associated with NPS occurrence, severe cohabitant caregiver burden is significantly associated with some NPS sub-symptoms (delusion, aggressiveness, emotional symptoms, and apathy or appetite loss) [52,53,56,58,59]. While caregiver burden, including physical distress and psychological frustrations for patients with dementia, is among causative factors leading to severe NPSs, these factors may also occur as a response to severe NPSs, creating a “distressful vicious circle” between the caregiver and the patient and disrupting a favorable interrelationship (Figure 3) [22]. To halt a “distressful vicious circle,” comprehensive non-pharmacological interventions (e.g., skill training for caregivers, education for caregivers, activity planning and environmental redesign, enhanced support for caregivers, and self-care techniques for caregivers) for cohabitant caregivers can contribute indirectly to a reduction in the severity of NPS in patients with dementia [60].

Among demographic factors, racial type and educational level may be interactive factors in economic matters connected to receiving care or medical services and financial support [52,55,56,61]. Background factors relevant to daily life and self-care support including medication compliance, which is related to adequate observation, can influence the subsequent response to treatment for NPSs [61]. Also, gender differences in patients with AD significantly influence the presence of a cohabitant and marital status (never married, widowed, divorced, and separated), which may be associated with various lifestyle changes, including solitude or social isolation, relevant to loneliness [54,55,56]. Especially, serious life events, including bereavement, may lead to loneliness or depressive symptoms in elderly widows [62]. Concerning the residential situation, if intrafamily psychological conflict causes severe caregiver burdens resulting in dysphoria in the patient, creating time or spatial distance between the patient and the caregiver through the use of a day care service or a short-stay service, including the relocation of the patient, may be necessary to reduce psychological distress in both parties [22,63,64].

Furthermore, these serious symptoms can merge with various other neuropsychiatric sub-symptoms (e.g., euphoria, apathy, disinhibition, repetitive behaviors, eating problems, and sleep disturbance), which may hinder clinicians from focusing on targeted symptoms [22,57]. Given that secondary delusions can originate from a guilty mind and pessimistic ideation can occur because of depressive symptoms, pharmacological treatments should prioritize antidepressants, rather than antipsychotics [22].

When such psychosocial, demographic, or comorbid sub-symptoms are not taken into consideration, patients might not respond to the continuous administration of pharmacological treatments for NPSs [22]. Excessive pharmacological interventions for NPSs without consideration of the psychopathological mechanisms may obscure the clinical situation, preventing clinicians from finding solutions. Therefore, unraveling the putative psychopathological mechanisms of complex symptoms may be helpful to clinicians engaged in designing a reasonable pharmacological strategy or planning personalized non-pharmacological interventions [10,11,12,13,22].

### 3.3. Neurocognitive Viewpoints: Neurocognitive Relevance, Longitudinal Cognitive Slope, and Self-Awareness

In reviewing the long-term degenerative course of AD, associations between neurocognitive factors and NPS occurrences have been discussed from the viewpoints of (1) primary neurocognitive relevance, (2) secondary neurocognitive relevance, and (3) poor self-awareness.

As an example of primary neurocognitive relevance, the severity and frequency of apathy have been positively correlated with global neurocognitive deterioration in patients with neurocognitive disorders [65,66,67]. The effectiveness of methylphenidate as a pharmacological treatment for apathy in patients with mild–moderate AD has been investigated in some studies [68,69,70], and significant efficacy and safety was found in a meta-analysis study [71]. However, the ineffectiveness of pharmacotherapy for the treatment of apathy in patients with severe AD is likely to be a direct result of neurocognitive deterioration [65,66,67].

As an example of secondary neurocognitive relevance, DMS in AD occurs when a patient believes that, for example, one’s house is not one’s home, that one’s spouse is an imposter, or that televised images or one’s mirror image are actually present in the house [72,73,74]. Beginning at an early disease stage, patients with AD have difficulty recalling recent episodic memories (recent autobiographical: RA), resulting in failures to update new episodic memories [75]. Meanwhile, preserved remote personal episodic (old autobiographical: OA) memory in patients with AD has been shown to persist despite progression to a severe dementia stage [76]. Thus, an imbalance between a lack of RA memory (RAM) and preserved OA memory (OAM) may cause distorted interpretations of objects present in reality, which patients with AD may then misidentify (for example, people, spaces, occupations) based on information retrieved from old memories [76]. If preserved OAM are activated by cognitive enhancers or emotional enhancements, patients with AD may believe that their actual daughter or their own mirror image is their mother or sister as a result of the misidentification of a familiar face (their daughter or their own face) with an old familiar face (their mother or sister) (Figure 4) [77,78].

Thirdly, rapid neurocognitive reduction through a certain duration (neurocognitive slope or gradient) has been significantly associated with some NPS sub-symptoms during the long-term course of AD, possibly disturbing conventional pharmacological treatments [4,79,80,81,82]. While rapid neurocognitive reduction may influence psychosis or affective symptoms, such symptoms are also inversely responsible for the cognitive and functional deterioration [4,79,80,81,82]. Slower cognitive progression in AD is associated with a better treatment response for NPSs over the long-term course [79]. We speculated that rapid cognitive reductions influence the discrepancy between the implicit world based on cognitive deterioration and explicit recognition of the real world. This produces affective confusion in daily life or causes the patient to jump to conclusions in the ideational process [83]. Therefore, the undesirable effects of psychotropics on rapid cognitive reductions may also make clinicians hesitant to determine long-term courses for psychotropics usage [14,15,23,84]. The observation of person DMS (e.g., Capgras syndrome; Fregoli syndrome; phantom border syndrome) in a patient with AD suggests the presence of complex neuropathological diseases involving AD and various other neurodegenerative diseases (e.g., synucleinopathy; prionopathy; amyloidopathy) [85,86]. DMS has been frequently reported in patients with AD and occurs during a moderately progressed stage of dementia (MMSE total score of about 13 points) [74]. Furthermore, as a distinct difference in cognitive profiles between DLB and AD, prominent deficits on neuropsychological tests of visuospatial ability and attention in patients with DLB have been previously discussed [87,88]. Thus, to determine each disease treatment strategy over a long-term course, clinicians should identify landmarks caused by differences between AD and DLB from a neurocognitive viewpoint.

Fourthly, self-awareness of cognitive symptoms and NPSs may influence the prognostic treatment response, since poor self-awareness can lead to refusal of care, including medication adherence, known as “anosognosia” [89,90,91] In patients with coping difficulties originating from anosognosia, comorbid NPSs can disturb the treatment intervention itself, delaying a solution. On the other hand, a preserved self-awareness of one’s cognitive reductions, including memory disorder, in patients with mild AD can cause pessimistic ideation concerning the future, leading to depressive symptoms and an extreme underestimation via metacognitive functions of one’s symptoms, potentially causing suicidal behavior [89,90,91]. A preserved self-awareness of one’s cognitive symptoms in patients with AD may require additional urgent pharmacological attention to prevent suicidal attempts over a longitudinal course of preventive interventions for neurodegenerative progression.

Collectively, neurocognitive progression as a core symptom in AD may induce subsequent neuropsychiatric problems via the following neurocognitive mechanisms of primary or secondary cognitive relevance and poor self-awareness: (1) functional reductions directly relevant to apathy; (2) indirect reactions via complex mechanisms resulting from compensative responses or activation (e.g., DMS, depression; agitation; aberrant motor behaviors). These mechanisms can also cause p-TRENS-AD if the above neurocognitive mechanisms are overlooked, obscuring the optimal treatment decisions that should be made.

## 4. Perspectives on Strategies for p-TRENS-AD 

Presently, interpersonal interventions (e.g., skills training for caregivers, education for caregivers, activity planning and environmental redesign, enhancing support for caregivers, and self-care techniques for caregivers) for cohabitant caregivers and non-pharmacological interventions (e.g., music therapy, reminiscence, and behavioral management techniques) against NPSs in patients with AD are warranted to reduce the severity of NPSs in dementia as a first-line approach [11,12,60,92]. However, if subsequent pharmacological approaches for NPSs are not improved, long-term treatment strategies based on the above mechanisms should be considered. In this section, we will discuss specific strategies for treating p-TRENS-AD with reference to previous studies examining pharmacological and non-pharmacological treatments for treatment-resistant mental disorders (Table 3).

### 4.1. Patient Relocation for the Treatment of p-TRENS-AD

Patient relocation into specialized care units that include appropriate facilities and specialized staff who can perform “behavioral management techniques” may be considered for patients with wandering or inappropriate behaviors for which optimal pharmacological treatments have not been determined applicable [11,22,27,93]. Some previous reports have shown that NPS in patients with dementia or in elderly people improved after a transfer to a special psychogeriatric ward for approximately 1.5–12 months of short or long-term hospitalization [63,94]. Haddad et al. has shown that admission to a more architecturally suitable facility for 12 weeks reduced delusion, euphoria, disinhibition, apathy, and agitation among NPS sub-symptoms in elderly people [64]. Therefore, relocation into a specialized care facility or unit may improve treatment-resistant NPSs in patients with AD [92].

### 4.2. Augmentation or Combination Therapy for Depression in p-TRENS-AD

Particularly for the treatment of depressive mood in patients with psychosis, the augmentation of antipsychotic therapy with an antidepressant medication may be helpful [51,95,96]. However, a recent network meta-analysis recommended that for patients with depression who are refractory to treatment with an initial antidepressant, the augmentation of antidepressant therapy with lower doses of an AAP (aripiprazole, brexpiprazole, cariprazine, olanzapine, quetiapine, or risperidone), dopamine compounds (lisdexamfetamine and modafinil), lithium, and thyroid hormone (liothyronine and T4) is feasible from the viewpoints of safety and efficacy [97]. However, to our knowledge, the duration of augmentation therapies has not been noted for elderly people [97]. The AVP-786 trial (dextromethorphan and quinidine) and the AXS-05 trial (dextromethorphan and bupropion) are ongoing investigations of drug efficacy in AD patients with agitation or depression [98,99,100]. It may be necessary for clinicians to decide between a target policy of recurrence prevention or accelerating antidepressant effects (boosting the effect during the early phase), and an exit strategy should be discussed in the future.

### 4.3. Neuromodulation for Depression and Apathy in p-TRENS-AD 

When pharmacotherapy is ineffective against apathy or depression, neuromodulation, including modified electric convulsive therapy, repetitive transcranial magnetic stimulation, and transcranial direct current stimulation, may be effective alternative strategic treatments for patients with AD [101,102,103]. However, repeated anodal transcranial direct current stimulation across the left dorsolateral prefrontal cortex had no effect on apathy in patients with AD; thus, future studies should explore noninvasive brain stimulation, such as repetitive transcranial magnetic stimulation, across the dorsolateral prefrontal cortex with the aim of targeting apathy symptoms in patients with AD [104].

### 4.4. Perspective as a Time-Dependent Solution 

A recent meta-analysis investigated the significant placebo effect on NPS that has been observed in patients with dementia who participated in some randomized controlled trials (RCTs) [105]. Some RCTs and cohort studies have also shown no difference in efficacy for the alleviation of NPS between active drugs and placebos over a long-term treatment course; however, the severity of the main NPS sub-symptoms (e.g., psychosis; agitation; depression) were almost halved [49,64,79,94,106]. In the CATIE-AD trial, the continuation of AAPs for the treatment of psychosis or agitation was revealed to be undesirable in subjects with AD; nevertheless, the overall NPS severity (Neuropsychiatric Inventory [NPI] score), including both the active and placebo drug groups, decreased to about 50% after 36 weeks [79]. Of the 77 subjects who completed the 36-week trial, the 21 patients in the placebo group experienced a decrease in the NPI score from 27.5 ± 17.1 to 11.6 ± 12.6 points (Figure 5) [107]. Likewise, two large RCTs investigating the effects of antidepressants on depression, anxiety, and agitation have shown reductions in agitation in the placebo groups to about 60–90% at 9 weeks and reductions in depression in the placebo groups to about 63% at 39 weeks [49,106]. In other words, the severity of NPS may decrease in a “time-dependent manner” regardless of medicine intake [107]. Similar to acute-phase treatments for patients with mental disorders, serious NPSs (e.g., major depression with suicidal ideation, psychosis causing harmful acting-out, agitation causing risk to self or others) in patients with dementia may require rapid solutions to alleviate urgent distress; however, hasty treatment decisions based on short-term goals can complicate or obscure long-term treatment. To avoid such pitfalls of long-term treatment, clinicians should consider the tendency for NPS severity to decrease over time. Collectively, these findings suggest that clinicians should keep their options for safe treatment open while considering the time-dependent pattern of NPS improvement and being attentive to early clinical predictors of future prognosis [108].

## 5. Discussion

As described above, the theoretical mechanisms in p-TRENS-AD can be divided into 3 main viewpoints: (1) biology; (2) psychosociology; and (3) neurocognition [22]. Regarding the biological viewpoint, pharmacokinetic and pharmacogenomic factors are directly relevant to clinical pharmacological effects in terms of the passage of drugs across the BBB or with regard to monoaminergic receptor availability and transmission. Metabolic factors of the CYP system are indirectly relevant to DDIs among prescribed medicines and psychotropics, with the potential to cause undesirable clinical outcomes, including a lack of efficacy or adverse effects. Neuropharmacological alterations in the monoaminergic or cholinergic system arising from consistent neurodegeneration and the aging process may also influence reciprocal pharmacodynamic actions via receptor availability or transmission in neurocircuits, leading to clinical treatment responses that differ in the elderly compared with those seen in younger patients. These pharmacological factors might also be influenced by functional alterations in drug response or intolerability arising from structural alterations caused by progressive neurodegeneration, either directly or indirectly. To resolve such neuropharmacological issues, a neuromodulation approach or tolerable combination therapies for p-TRENS-AD may be useful for both clinicians and caregivers. Next, each psychosocial or demographic factor associated with NPS emergence may complicate the origin of distressful vicious circles that can lead to treatment resistance. In particular, the significant association between NPS severity and caregiver burden may exacerbate symptoms, leading to the development of p-TRENS-AD. Moreover, ignoring comorbid sub-symptoms can lead to delays in the start of interventions for the true causative factor, the resolution of which should be prioritized. Therefore, noticing critical interpersonal psychopathological mechanisms of NPS in AD in a timely manner may contribute to appropriate treatments (e.g., relocation to a specialized care unit), preventing progression toward p-TRENS-AD. Thirdly, neurocognitive impairment may contribute to alterations in interpretations of the present situation, which can cause secondary affective reactions or actions based on misidentified convictions. For instance, preserved self-awareness, autobiographical memory imbalance, and rapid cognitive reductions in patients with AD may indirectly cause redundant pessimistic ideation connected to dysphoric feelings and misidentification or confusion reflecting a distorted interpretation of present objects. On the other hand, when sub-symptoms do not improve despite repeat trials of conventional pharmacological treatments, progressive neurodegenerative changes may lead to a direct overlap of these sub-symptoms with apathy. Therefore, considering the association between the main cognitive impairment and NPSs may contribute to the prevention or modification of worsened NPSs in long-term care plans. The need to treat apathy arising from a terminal stage in a consistently progressive dementia course has been discussed [109]. Additional psychosocial approaches and counseling for patients with preserved self-awareness of their own clinical progressive symptoms and psychoeducation for caregivers might be needed, rather than pharmacological interventions, beginning at an early stage of the AD clinical course.

Innovative disease-modifying therapies (DMT) and drug re-positioning therapies targeting causative proteins (*β*-amyloid; tau) among radical cure treatments are presently being investigated in some ongoing trials, but disease progression to a terminal stage unfortunately cannot be completely stopped [110]. Clinicians should change their strategic plans at each phase of the chronic course, shifting their targeted outcomes from preventing disease progression during the early stage to prolonging QOL during the terminal stage [110,111]. In terms of comprehensive coping strategies for end-of-life care in patients with long-term dementia, clinicians need to take a humanistic approach that will support patients, caregivers, and families, so as to modify the end-of-life prognosis of AD in the future [110,111]. The ultimate treatment target in people with dementia has recently shifted from prolonging lifespan through preventive lifestyle to promoting well-being QOL, according to the World Health Organization (WHO)’s proposed agenda, from the viewpoint of the dignity of human rights [111]. When treatments, including pharmacological approaches, are forced onto patients against their will, there may be a risk of human rights violation. Consequently, the need for treatment should be considered together with the patient and the patient’s family, arriving at a decision together. The advocated human right-based approach (RBA) has been in the spotlight as a common universal philosophy for treating people with dementia, but discrepancies between the ideal approach and real-world care strategies were found in an RCT investigating the effectiveness of RBA for QOL in dementia [112]. The efficacy of pharmacological treatments for NPSs as prognostic predictors may be comparatively heterogeneous, as described above, suggesting that clinicians should provide personalized care, including non-pharmacological interventions, using a “time-dependent solution” as a next step. Therefore, to improve the QOL of people with AD using RBA, understanding the distress mechanism in people with dementia who are suffering from NPSs in terms of a “time-dependent solution” may be beneficial to both clinicians and caregivers, even in the absence of consistent methods of improvement including DMTs and replacement therapy.

## 6. Conclusions

NPSs in patients with AD are often difficult to treat pharmacologically, which frequently leads to refractory symptoms (e.g., a need to switch psychotropics despite an adequate dose and an adequate treatment duration). The effectiveness of DMT for AD has not yet been consistently proven, suggesting that clinicians should adopt personalized care strategies based on a “time-dependent solution”. To enable an improved prognosis linked to well-being QOL in the future, further elucidation of AD pathogenesis and comprehensive solutions for pharmacological treatment-resistant NPSs based on the pathophysiological viewpoints of (1) biology, (2) psychosociology, and (3) neurocognition is needed.

## Figures and Tables

**Figure 1 jpm-12-01365-f001:**
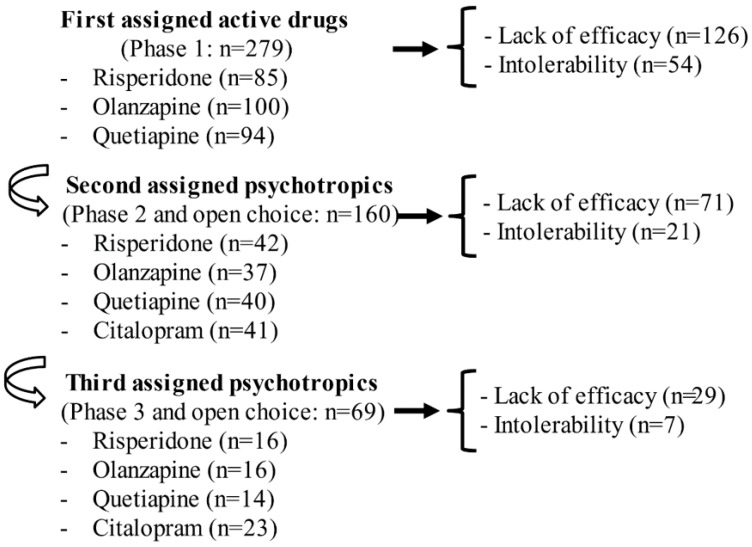
Switching of AAPs or citalopram in the CATIE-AD trial [15]. The results are summarized based on reference [15] and a re-analysis of the raw data. **Abbreviations:** AAPs: atypical antipsychotics, CATIE-AD: Clinical Antipsychotic Trials of Intervention Effectiveness–Alzheimer’s Disease.

**Figure 2 jpm-12-01365-f002:**
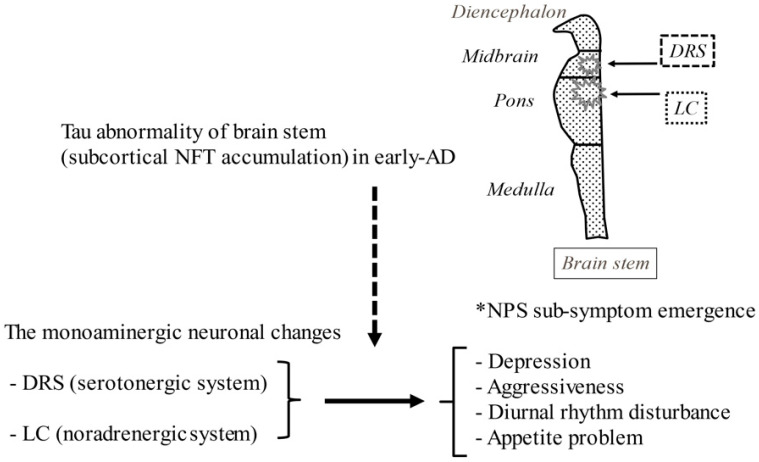
Some NPSs in early-AD may originate from the involvement of monoaminergic neurons, including the serotonergic DRS and the noradrenergic LC. This might be relevant to early tau protein abnormalities, suggesting subcortical NFT accumulation. **Abbreviations:** AD: Alzheimer’s Disease, DRS: dorsal raphe nucleus, LC: locus coeruleus, NFT: neurofibrillary tangle, NPS: neuropsychiatric symptom.

**Figure 3 jpm-12-01365-f003:**
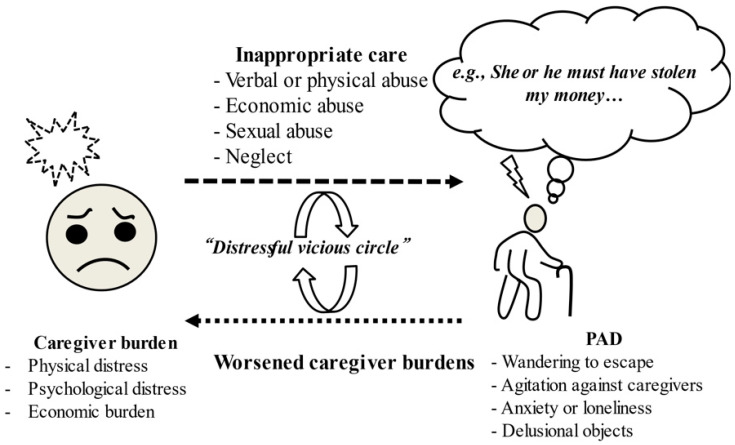
Caregiver burden and NPS interact to create a “*distressful vicious circle*” between the caregiver and the PAD, causing a deterioration in their interpersonal relationship. While caregiver burden, including physical distress and psychological frustrations arising from the care of dementia patients, can be causative factors leading to severe NPSs, these factors may also occur as a response to severe NPSs, creating a “distressful vicious circle” between the caregiver and the patient and disrupting favorable interrelationships. Abbreviations: NPSs: neuropsychiatric symptoms, PAD: patient with Alzheimer’s disease.

**Figure 4 jpm-12-01365-f004:**
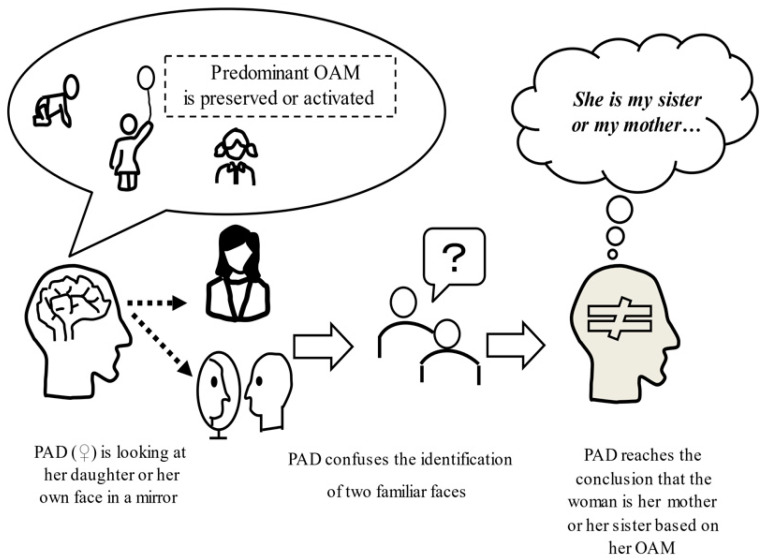
The predominance of OAM can cause delusional misidentifications. If preserved OAM are activated by cognitive enhancers or emotional enhancements, patients with AD may believe that their actual daughter or their own mirror image is their mother or sister as a result of the misidentification of a familiar face (their daughter or their own face) with an old familiar face (their mother or sister). **Abbreviations**: OAM: old-autobiographical memory, PAD: patient with Alzheimer’s disease.

**Figure 5 jpm-12-01365-f005:**
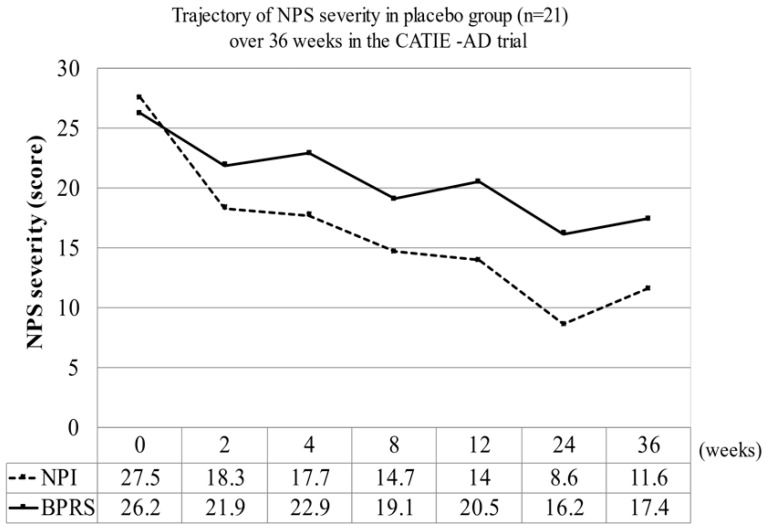
In the CATIE-AD trial, the NPS severity (BPRS or NPI scores) decreased in a placebo group to about 40–60% after 36 weeks. [108] Among the 77 subjects who completed the 36-week CATIE-AD trial, the NPI scores of the 21 patients in the placebo group decreased from 27.5 ± 17.1 to 11.6 ± 12.6 points, while the BPRS scores decreased from 26.2 ± 14.5 to 17.4 ± 114.1 points. **Abbreviations**: BPRS: Brief Psychiatric Rating Scale, CATIE-AD: Clinical Antipsychotic Trials of Intervention Effectiveness–Alzheimer’s Disease, NPI: Neuropsychiatric Inventory, NPS: neuropsychiatric symptom.

**Table 1 jpm-12-01365-t001:** Summary of recommended pharmacological treatments for NPSs in patients with Alzheimer’s disease [22].

Medication	Target Symptoms	Major Adverse Effects	Comments
**AAP**			
: Risperidone, Olanzapine, Quetiapine, Aripiprazole	Psychosis; Agitation	Cerebrovascular adverse events, pneumonia, metabolic syndromes, falls, cognitive declines	Since psychosis or agitation can cause harm to the patient or others, AAP should be prioritized for a limited duration.
**Antidepressant**			
: SSRIs (sertraline, citalopram)	Agitation	(1) SSRI: prolongation of the QTc interval on ECG, cognitive declines, gastrointestinal symptoms (2) Trazodone: daytime somnolence, postural hypotension	Agitation may require treatment with a low dose of SSRI, while trazodone may play a role as a sleep modulator.
: Trazodone	Sleep disturbance		
**Anticonvulsant** : Carbamazepine	Agitation	Hematotoxicity, interaction against other drugs, falls, impairment of balance	AAPs treatment-resistant agitation can be treated with carbamazepine.
**Anti-dementia drug** : Memantine	Psychosis, Agitation,	Headache, constipation, dizziness	Little evidence supports the effectiveness of memantine for agitation and psychosis.

**Abbreviations:** AAP: Atypical antipsychotic, ECG: electrocardiogram, NPSs: neuropsychiatry symptoms, SSRI: selective serotonin reuptake inhibitor.

**Table 2 jpm-12-01365-t002:** Summary of p-TRENS-AD.

Causative Factors of p-TRENS-AD	Comments
**(1) Biological factors**: - pharmacokinetics or metabolic (DDI) change - pharmacogenomics - neuropharmacological relevance	- Monoamine associate gene polymorphisms and DDI via CYP system or BBB penetration contributes to treatment response or intolerability. Neuropharmacological alterations based on structural degeneration are relevant to pharmacological functions (e.g., monoaminergic-cholinergic system alterations).
**(2) Psychosocial factors**: - severe caregiver burden - demographic factors as unchangeable background - multitargeted sub-symptoms	- Severe NPSs and severe caregiver burden cause a “distressful vicious circle” in PAD and cohabitant. The serious symptoms merge with various other neuropsychiatric sub-symptoms, which may hinder clinicians from focusing on targeted symptoms.
**(3) Neurocognitive factors**: - rapid cognitive reduction - OAM predominance - poor self-awareness (anosognosia)	- Rapid cognitive slope due to psychotropics, OAM activation based on antidementia drugs, and poor self-unawareness may cause delusional misidentification, confusion reflecting a distorted interpretation of present objects, or pessimistic ideation connected to dysphoric feelings.

**Abbreviations**: BBB: blood-brain barrier, CYP: cytochrome P450, DDI: drug–drug interactions, NPSs: neuropsychiatric symptoms, OAM: old-autobiographical memory, PAD: patient with Alzheimer’s disease.

**Table 3 jpm-12-01365-t003:** Perspectives on treatment strategies for p-TRENS-AD.

Future or Present Treatment Strategies	Comments
**(1) Patient relocation for treatment**	Patient relocation into specialized care units that include appropriate facilities and specialized staff who can perform “behavioral management techniques” may be considered for patients with wandering or inappropriate behaviors for which optimal pharmacological treatments have not been determined.
**(2) Augmentation or combination therapy for depression**	Antidepressant therapy is augmented with low doses of an AAP, dopamine compounds, lithium, or thyroid hormone. The AVP-786 trial (dextromethorphan and quinidine) and the AXS-05 trial (dextromethorphan and bupropion) are ongoing investigations of treatment efficacy in AD patients with agitation or depression.
**(3) Neuromodulation for depression and apathy**	Neuromodulation including modified electric convulsive therapy, repetitive transcranial magnetic stimulation, and transcranial direct current stimulation may be effective alternative treatments for depression/apathy in AD.
**(4) Time-dependent solution**	Clinicians should select safe treatment options bearing this time-dependent pattern of NPS improvement in mind and being attentive to early clinical predictors of future prognosis.

**Abbreviations:** AAP: atypical antipsychotics, AD: Alzheimer’s disease, NPS: neuropsychiatric symptom.

## Data Availability

Not applicable.

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
