# Peer review of "Pathogenesis and Personalized Interventions for Pharmacological Treatment-Resistant Neuropsychiatric Symptoms in Alzheimer’s Disease"

_jpm, 2022, doi:10.3390/jpm12091365_

Round 1
Reviewer 1 Report (Previous Reviewer 1)
I have read the manuscript entitled “Pathogenesis and personalized interventions for pharmacological treatment-resistant neuropsychiatric symptoms in Alzheimer’s disease” submitted to the Journal of Personalized Medicine.
This manuscript by Tomoyuki Nagata and colleagues discusses on the difficulties in the pharmacological treatment of elderly patients with AD. The authors advocate the NPSs in patients with AD who have treatment resistance as p-TRENS-AD and discuss specific strategies for treating p-TRENS-AD. The manuscript is well-written containing extremely important information for the coming super-aged society.
Minor points
1) In the section1 line 78-83, the authors mentioned the clinical trials termed “CATIE-AD” and discussed about this trial in the following sentences. However, it is not easy to get the point since the aims and details for CATIE-AD are unclear in this manuscript although the authors referenced the report. I think that the brief description of the CATIE-AD is needed before starting the discussion.
2) I think that the addition of diagrams will help readers understand this review better. I recommend adding diagrams for section 3.1 and section 4.
3) I think that figure legends should be more improved. Currently, it is hard to understand the meaning of figures.
Author Response
Reply to comments by Referee 1
Thank you very much for your comments and useful suggestions. We have revised our manuscript accordingly.
Comment #1:
In the section1 line 78-83, the authors mentioned the clinical trials termed “CATIE-AD”
and discussed about this trial in the following sentences. However, it is not easy to get the point since the aims and details for CATIE-AD are unclear in this manuscript although the authors referenced the report. I think that the brief description of the CATIE-AD is needed before starting the discussion.
Reply to Comment #1:
Thank you for your thoughtful comments. We agree with the reviewer’s point. To avoid confusion, we have added a detailed description of “CATIE-AD” to section 1.2 of the Introduction.
Added or changed sentence:
Antipsychotics have conventionally been presumed to be more effective than other medicines for improving psychosis or agitation in patients with AD; however, the reproducibility of such results is unclear [11-13]. The Clinical Antipsychotic Trials of Intervention Effectiveness–Alzheimer’s Disease (CATIE-AD) was performed from 2000 to 2004. This 36-week, large-scale, double-blind, placebo-controlled study investigated the longitudinal effectiveness and safety of atypical antipsychotics (AAP) for the treatment of mainly psychotic or aggressive symptoms in 421 patients with AD.
Comment #2:
I think that the addition of diagrams will help readers understand this review better. I recommend adding diagrams for section 3.1 and section 4.
Reply to Comment #2:
Thank you for your comments. We agree with the reviewer’s point completely. Overall, to avoid confusion, we have emphasized the summarized ‘Table 2’ in each section heading title (3-1, 2, and 3). Moreover, we have added one figure to section 3.1 and one table to section 4.1, according to the reviewer’s advice.
Comment #3:
I think that figure legends should be more improved. Currently, it is hard to understand the meaning of figures.
Reply to Comment #3:
In response to the reviewer’s suggestion, we have emphasized the details of each Figure Legend.
Reviewer 2 Report (New Reviewer)
This narrative review considers the pathogenesis of neuropsychiatric symptoms in dementia that are difficult to treat pharmacologically and presents perspectives on their treatment. They review pharmacogenetic, psychosocial and neurocognitive aspects.
P5 line 198-209: the authors try to define treatment resistant NPS in AD and get into a circle argumentation because they choose the pTRANS-AD for treatments that per se are defined as difficult to treat pharmacologically, but can be treated with multiple changes, and those sub-symptoms that do not respond at all. Authors should only use the term p-TRANS-AD for the latter.
P7 line 309-310: mentioning of the LC and the raphe should be extended tot he importance of sleep factors which are regulated by these centers. The p-TRANS-AD also should include some aspects of circadian regulation which may cause sundowning, aggressiveness etc. The focus of the biological aspects is very much on depression and psychosis missing other symptoms like delusions, hallucinations etc. These symptoms should also be covered with some literature even if it is scarce.
The therapeutic proposals are well done concerning pharmacological and psychosocial interventions. A recommendation or proposal should be done in a differential way for acute and subacute NPS which is only marginally addressed on p 12 line 579.
Overall, this is a well written interesting paper that reviews NPS in dementia and which tries to establish recommendations for NPS in demented persons.
Author Response
Reply to comments by Referee 2
Thank you very much for your comments and useful suggestions. We have revised our manuscript accordingly.
Comment #1:
P5 line 198-209: the authors try to define treatment resistant NPS in AD and get into a circle argumentation because they choose the pTRANS-AD for treatments that per se are defined as difficult to treat pharmacologically, but can be treated with multiple changes, and those sub-symptoms that do not respond at all. Authors should only use the term p-TRANS-AD for the latter.
Reply to Comment #1:
Thank you for your thoughtful comments regarding ‘circle argumentation’ in the definition of treatment-resistance. In the present review, ‘specific issues’ regarding pharmacological treatment difficulties in the elderly or people with dementia can be emphasized, unlike the situation for patients with schizophrenia or mood disorders. However, the pharmacological treatment efficacy in people with dementia has been limited; therefore, we have focused on only urgent NPS sub-symptoms (psychosis, aggressiveness, and depression) requiring pharmacological treatment in the present review. However, the effectiveness of pharmacological treatment has been discussed in previous studies, but sub-symptoms such as wandering, perseverative shouting, and some sexually inappropriate behaviors are unlikely to respond to a pharmacological approach. These sub-symptoms may be ‘not applicable for a pharmacological approach’ from their initial appearance. The p-TRENS-AD overlapping with such sub-symptoms (wandering, perseverative shouting, and inappropriate behaviors) may also need to be discussed as a critical matter requiring resolution. We have re-emphasized this content in the second section.
Re-emphasized sentences:
The effectiveness of pharmacological treatment has been discussed in previous studies, but sub-symptoms such as wandering, perseverative shouting, and some sexually inappropriate behaviors are unlikely to respond to a pharmacological approach [11,22,27]. Thus, a pharmacological approach might not be suitable for the treatment of these sub-symptoms when they first appear. The p-TRENS-AD overlapping with such sub-symptoms (wandering, perseverative shouting, and inappropriate behaviors) should be defined as a solution to a difficult issue.
Comment #2:
P7 line 309-310: mentioning of the LC and the raphe should be extended to the importance of sleep factors which are regulated by these centers. The p-TRANS-AD also should include some aspects of circadian regulation which may cause sundowning, aggressiveness etc. The focus of the biological aspects is very much on depression and psychosis missing other symptoms like delusions, hallucinations etc. These symptoms should also be covered with some literature even if it is scarce.
Reply to Comment #2:
Thank you for your thoughtful comments. We have added a description regarding ‘diurnal rhythm disturbances’ to section 3.1.
Added or changed sentence:
The DRS or LC is also involved in the control of the sleep-wake cycle, which means that its impairment may cause diurnal (circadian) rhythm disorders including wake-sleep cycle disturbances in patients with AD [46,47].
Comment #3:
The therapeutic proposals are well done concerning pharmacological and psychosocial interventions. A recommendation or proposal should be done in a differential way for acute and subacute NPS which is only marginally addressed on p 12 line 579.
Reply to Comment #3:
Thank you for your comments. When serious NPSs (e.g., major depression with suicidal ideation, psychosis causing harmful acting-out, agitation causing risk to self or others) in patients with dementia are regarded as acute mental disorders, clinicians should undertake a pharmacological strategy while keeping their options for long-term treatment open. Simultaneously, a viewpoint of ‘time-dependent pattern of NPS improvement’ is needed to avoid the pit-falls of p-TRENS; this content has been described in section 4.4.
Revised sentence:
Similar to acute-phase treatments for patients with mental disorders, serious NPSs (e.g., major depression with suicidal ideation, psychosis causing harmful acting-out, agitation causing risk to self or others) in patients with dementia may require rapid solutions to alleviate urgent distress; however, hasty treatment decisions based on short-term goals can complicate or obscure long-term treatment. To avoid such pitfalls of long-term treatment, clinicians should consider the tendency for NPS severity to decrease over time. Collectively, these findings suggest that clinicians should keep their options for safe treatment open while considering the time-dependent pattern of NPS improvement and being attentive to early clinical predictors of future prognosis [109].
Comment #4:
Overall, this is a well written interesting paper that reviews NPS in dementia and which tries to establish recommendations for NPS in demented persons.
Reply to Comment #4:
Thank you for your comments.
This manuscript is a resubmission of an earlier submission. The following is a list of the peer review reports and author responses from that submission.
Round 1
Reviewer 1 Report
I have read the manuscript (ID: jpm-1791068) entitled “Pathogenesis and personalized interventions for pharmacological treatment-resistant neuropsychiatric symptoms in Alzheimer’s disease” submitted to the Journal of Personalized Medicine.
This manuscript by Tomoyuki Nagata and colleagues discusses on the difficulties in the pharmacological treatment of elderly patients with AD. The authors advocate the NPSs in patients with AD who have treatment resistance as p-TRENS-AD and discuss specific strategies for treating p-TRENS-AD. The manuscript is well-written containing extremely important information for the coming super-aged society. The publication would be appropriate after minor revision.
Minor points
1) In the section1 line 78-83, the authors mentioned the clinical trials termed “CATIE-AD” and discussed about this trial in the following sentences. However, it is not easy to get the point since the aims and details for CATIE-AD are unclear in this manuscript although the authors referenced the report. I think that the brief description of the CATIE-AD is needed before starting the discussion.
2) I think that the addition of diagrams will help readers understand this review better. I recommend adding diagrams for section 3.1 and section 4.
3) I think that figure legends should be more improved. Currently, it is hard to understand the meaning of figures.